# The In Silico Characterization of Monocotyledonous α-l-Arabinofuranosidases on the Example of Maize

**DOI:** 10.3390/life13020266

**Published:** 2023-01-18

**Authors:** Alsu Nazipova, Olga Makshakova, Liudmila Kozlova

**Affiliations:** 1The Laboratory of Plant Cell Growth Mechanisms, Kazan Institute of Biochemistry and Biophysics FRC Kazan Scientific Center of RAS, Lobachevsky Str. 2/31, 420111 Kazan, Russia; 2The Laboratory of Biophysical Chemistry of Nanosystems, Kazan Institute of Biochemistry and Biophysics FRC Kazan Scientific Center of RAS, Lobachevsky Str. 2/31, 420111 Kazan, Russia

**Keywords:** α-l-arabinofuranosidase, arabinoxylan, maize, homology modeling, molecular docking

## Abstract

Plant α-l-arabinofuranosidases remove terminal arabinose from arabinose-containing substrates such as plant cell wall polysaccharides, including arabinoxylans, arabinogalactans, and arabinans. In plants, de-arabinosylation of cell wall polysaccharides accompanies different physiological processes such as fruit ripening and elongation growth. In this report, we address the diversity of plant α-l-arabinofuranosidases of the glycoside hydrolase (GH) family 51 through their phylogenetic analysis as well as their structural features. The CBM4-like domain at N-terminus was found to exist only in GH51 family proteins and was detected in almost 90% of plant sequences. This domain is similar to bacterial CBM4, but due to substitutions of key amino acid residues, it does not appear to be able to bind carbohydrates. Despite isoenzymes of GH51 being abundant, in particular in cereals, almost half of the GH51 proteins in *Poales* have a mutation of the acid/base residue in the catalytic site, making them potentially inactive. Open-source data on the transcription and translation of GH51 isoforms in maize were analyzed to discuss possible functions of individual isoenzymes. The results of homology modeling and molecular docking showed that the substrate binding site can accurately accommodate terminal arabinofuranose and that arabinoxylan is a more favorable ligand for all maize GH51 enzymes than arabinan.

## 1. Introduction

α-l-Arabinofuranosidases (EC 3.2.1.55) are exo-active glycoside hydrolases (GHs) that remove α-l-arabinofuranose from the non-reducing end of arabinose-containing substrates. The α-l-arabinofuranosidases have been classified into eight GH families: 2, 3, 10, 43, 51, 54, 62, and 159 [1]. Bacteria and fungi use α-l-arabinofuranosidases to degrade plant cell wall polysaccharides such as arabinoxylan, arabinan, arabinogalactan, and their oligosaccharides [2,3]. These enzymes are also used in cocktails to convert plant biomass for different industrial purposes [4], making their investigation more important from a practical point of view.

In plants, α-l-arabinofuranosidases are known to be involved in modification/degradation of cell wall polysaccharides during various physiological processes such as seed storage mobilization [5], fruit ripening [6,7] and fruit softening [8,9], turnover of arabinogalactan proteins during seed maturation [10], and in the proper release of seed mucilage from secretory cells [11]. In monocotyledons (*Liliopsida*), α-l-arabinofuranosidase activity is increased during cessation of elongation growth and presumably contributes to the arabinoxylan debranching [12,13,14]. The latter is important for maintaining the correct architecture of plant cell walls, especially primary cell walls of type II, where arabinoxylan serves as the main hemicellulose [15].

In higher plants, enzymes with α-l-arabinofuranosidase activity belong to GH families 3 and 51 [16]. However, members of GH3 demonstrate β-d-glucosidase (EC 3.2.1.21), β-d-xylosidase (EC 3.2.1.37) or bifunctional β-d-xylosidase/α-l-arabinofuranosidase activities in dicotyledons [11,17,18] and monocotyledons [19], whereas GH51 enzymes exhibit mostly an α-l-arabinofuranosidase activity with a single example of a bifunctional α-l-arabinofuranosidase/β-d-xylosidase enzyme [17,20,21]. Some bacterial GH51 enzymes possess endoglucanase (EC 3.2.1.4), xylanase (EC 3.2.1.8), and cellobiohydrolase (EC 3.2.1.91) activities [1,22,23,24], but no such reports have yet been made for eukaryotes. For simplicity, we will refer to the plant sequences belonging to GH51 as α-l-arabinofuranosidases, as this is most likely, but it should be considered that this may not be true for all the sequences.

The GH51 enzymes are retaining glycosidases that preserve the anomeric configuration of the substrate after hydrolysis. Like most retaining enzymes, they use two glutamic acids as an acid/base and a nucleophile [25]. GH51 enzymes belong to clan A of GHs, so their active site is located at the N-terminal (β/α)_8_-barrel. Typical enzymes also possess a C-terminal β-sandwich domain that stabilizes the overall structure [25,26,27,28,29,30]. However, a recent study of the fungal enzyme MgGH51 showed the presence of an N-terminal domain similar to the carbohydrate-binding module (CBM) of family 4 with unknown function in addition to the aforementioned domains [31].

Plant members of the GH51 family do not have resolved crystal structures, but some have been studied biochemically. The barley enzyme HvAXAH1 removes arabinose from mono- and di-substituted arabinoxylans at high and low rates, respectively, utilizing sugar beet arabinan and larch wood arabinogalactan, but the latter at a low catalytic rate [21]. Rice enzymes OsARAF1 and OsARAF3 are involved in the degradation of primary cell wall arabinoxylans, which was revealed in mutant plants overexpressing the corresponding genes [32]. In the dicotyledonous plant *Arabidopsis thaliana*, the α-l-arabinofuranosidase AtARAF1 is involved in the modification of pectic arabinans as shown in overexpressing and knock-out mutant plants [33,34].

We previously identified eight genes in the maize genome as encoding GH51 α-l-arabinofuranosidases (*ZmaABF1-8*) and showed that the expression of most of them increases with the development of primary maize root cells, as well as the total α-l-arabinofuranosidase activity [14]. However, the *ZmaABF2* gene showed the opposite pattern of gene expression. It was up-regulated in the root cap and meristem [14]. A recent transcriptomic study confirmed this pattern and additionally pointed to co-expression of *ZmaABF1* with genes of secondary cell wall cellulose synthases, indicating the possible involvement of this enzyme in secondary cell wall formation [35]. Cereals have a notably evolved GH51 family [34], but neither the specific targets for these enzymes nor their physiological role are fully understood.

In this study, we re-predicted maize genes encoding putative α-l-arabinofuranosidases of the GH51 family in the most recent to date version of the maize genome. Transcription and translation data for these genes in different organs and tissues of maize were analyzed. Homology modeling using the resolved structures of α-l-arabinofuranosidases as templates and molecular docking experiments were performed to establish substrate preferences for maize GH51 enzymes. Phylogenetic analysis established the evolution of α-l-arabinofuranosidases in plants, revealing a widespread potentially deactivating mutation in the enzymes of some orders of monocotyledonous plants. A domain resembling the carbohydrate binding module that is specific exclusively to GH51 was revealed and its ability to interact with carbohydrates was discussed.

## 2. Materials and Methods

### 2.1. Identification of GH51 Proteins in Different Plant Species

The sequences of plant α-l-arabinofuranosidases were recognized in the genomes and transcriptomes of various plants in three ways: (1) searching for PF06964 (Alpha-L-AF_C) Pfam domain name in the Phytozome database v13 (https://phytozome-next.jgi.doe.gov/; [36]) (accessed on 1 April 2022); (2) Basic Local Alignment Search Tool (BLAST) search in the One Thousand Plant Transcriptome Project database (1KP, https://db.cngb.org/onekp/species/; [37]) (accessed on 20 April 2022) using AtARAF1 of *A. thaliana* as a query; (3) local searching for the PF06964 domain in preliminarily downloaded genomes using the *hmmsearch* function in the HMMer v3.3 [38]. The domain organization of the sequences found was verified using the InterProScan (https://www.ebi.ac.uk/interpro/search/sequence/; [39]) (accessed on 20 April 2022) and/or the web-based tool Batch-CD-search (https://www.ncbi.nlm.nih.gov/Structure/bwrpsb/bwrpsb.cgi; [40]) (accessed on 20 April 2022).

The PF06964 domain profile was downloaded from the Pfam 32.0 database (http://pfam.xfam.org/; [41]) (accessed on 19 April 2022). Plant genomes were downloaded from the EnsemblPlants (http://plants.ensembl.org/index.html; [42]) and GenBank (https://www.ncbi.nlm.nih.gov/protein/; [43]) databases (accessed on 16 April 2022); *Klebsormidium nitens* alga genome was downloaded from the *K. nitens* NIES-2285 genome project (http://www.plantmorphogenesis.bio.titech.ac.jp/~algae_genome_project/klebsormidium/index.html; [44]) (accessed on 16 April 2022), and plant transcriptomes were downloaded from the 1KP database. The resulting plant GH51 protein sequences were aligned using the Muscle algorithm (https://www.ebi.ac.uk/Tools/msa/muscle/; [45]) (accessed on 21 April 2022). Sequences less than 350 residues and/or not containing two conserved G-N-E/Q and S-E-Y-x-V motifs simultaneously were excluded from further analysis. A total of 139 out of 156 sequences remained after this filtering.

Three protein sequences (M8CET9, M8CFW8 of *Aegilops tauschii,* and 56386 of *Coccomyxa subellipsoidea* (listed in Appendix A)) were re-predicted in the web tools Augustus (https://bioinf.uni-greifswald.de/augustus/; [46]) and FGENESH (http://www.softberry.com/berry.phtml?topic=fgenesh&group=programs&subgroup=gfind; [47]) (accessed on 21 April 2022).

The maize genes encoding GH51 proteins were identified as described above in the maize B73 genome downloaded from the GenBank (Zm-B73-REFERENCE-NAM-5.0 assembly). The genes encoding GH51 α-l-arabinofuranosidase in the RefGen B73 AGPv3 5b+ genome assembly (downloaded from the Maize Genome database (MaizeGDB, https://www.maizegdb.org/; [48]) (accessed on 2 March 2022) were aligned to the Zm-B73-REFERENCE-NAM-5.0 genome assembly using a BLAST search of translated nucleotide databases using a protein query (https://blast.ncbi.nlm.nih.gov/Blast.cgi; [49]) (accessed on 3 March 2022). 

The presence of signal peptide in the maize sequences was predicted using the web tool SignalP 6.0 (https://services.healthtech.dtu.dk/service.php?SignalP; [50]) (accessed on 5 March 2022).

### 2.2. GH51 Gene and Protein Expression in Maize

Data on GH51 gene expression in different maize organs were obtained by Walley et al. [51] and Hoopes et al. [52] and downloaded from the MaizeGDB database. The proteome data were retrieved from the study by Walley et al. [51].

### 2.3. CBM4-like Domain Analysis

To create a CBM4-like domain profile, 11 bacterial, 16 fungal, and 111 plant GH51 sequences (listed in Appendix A) were used. The random full-length GH51 sequences of bacteria and fungi were obtained by protein BLAST search using the fragment (1–238 residues from the N-terminal end) of AtARAF1 as a query. Several sequences listed in Lagaert et al.’s (2014) review [2] and the sequence of MgGH51 were also included. The plant sequences were obtained as described above. The domain organization of all sequences was verified using InterProScan. The CBM4-like domain was established between two conserved L-x-V-D and S-x-x-P motifs (Appendix A) in the multiple sequence alignment (Muscle algorithm) of all obtained sequences. The parts of the alignment following the S-x-x-P motif were removed. The truncated alignment was converted into a Pfam domain profile using the *hmmbuild* function in HMMer-3.3.

To estimate the proportion of bacterial, fungal, insect, and plant GH51 proteins which contain a CBM4-like domain, the search of the generated domain profile using the *hmmscan* function (HMMer-3.3.) was performed. The protein sequences contained either the PF06964 domain or both the PF02018 and PF06964 domains, which were downloaded from the Uniprot (https://www.uniprot.org/; [53]) and Phytozome (https://phytozome-next.jgi.doe.gov/; [36]) databases (accessed on 17 May 2022). The sequences longer than 200 residues were used for search. The GH51 protein sequences of insects were found by the BLASTP search of the fragment (1–238 residues from N-terminal end) of AtARAF1 as a query in eukaryotes excluding fungi and plants.

The multiple sequence alignment was followed by a structural superposition determined by super method in the PyMOL Molecular Graphics System v2.2.0 [54]. All spatial structures were visualized in PyMOL.

The multiple sequence alignment of bacterial CBMs of family 4 and CBM4-like domains of MgGH51 and some plant GH51 proteins was performed using the Guidance 2 server with default parameters (http://guidance.tau.ac.il/; [55]) (accessed on 17 May 2022).

### 2.4. Phylogenetic Analysis

Sequences were aligned using the Muscle algorithm and the resulting alignments were subjected to phylogenetic analysis in IQ-TREE v2.1.3 [56]. The best-fit model of sequence evolution was computed using Model Finder (IQ-TREE) [57] and selected using a Bayesian Information Criterion. To construct phylogenetic dendrograms, an ultra-fast bootstrap branch support based on 10,000 replicates [58] was used. Trees were visualized in the iTOL v6.5.3 web service (https://itol.embl.de/; [59]) (accessed on 26 April and 17 May 2022). Abbreviated names of genes and proteins were used; full names are listed in Appendix A. Phylogenetic analysis of plant sequences of monocotyledons was preceded by the removal of ambiguous regions of multiple alignments using Gblocks v0.91b [60] with the following parameters: the maximum number of contiguous unconserved positions was 15; the minimum block length was 2; the allowed gap position was set as half (only positions where 50% or more of the sequences have a gap are treated as a gap position [60]).

### 2.5. Homology Models of GH51 Proteins

The crystal structure of MgGH51 from *Meripilus giganteus* co-crystalized with arabinose (PDB ID: 6zpy) was used as a template for homology modeling of plant proteins in MODELLER v. 9.15 [61]. The arabinose ligand was copied to the resultant models from the template to shape the highly conservative binding site of the model as in X-ray structure. The protein sequences of GH51 proteins from maize and AtARAF1 from *A. thaliana* were obtained as described above. The sequence and structure of MgGH51 and Tx-Abf from *Thermobacillus xylanilyticus* (PDB ID: 2vrq) were obtained from the Protein Data Bank (PDB, http://www.rcsb.org/; [62]) (accessed on 2 April 2022). Protein sequence pairwise alignment was performed in the ClustalW service (https://www.ebi.ac.uk/Tools/msa/clustalo/; [63]) (accessed on 2 April 2022). The 36, 74, 36, and 42 amino acid residues from the N-terminal end of ZmaABF1, ZmaABF2, ZmaABF5, and AtARAF1, respectively, were left out during homology modeling due to the presence of signal peptides (ZmaABF1, ZmaABF5, and AtARAF1) and the absence of a homologous fragment in the MgGH51 sequence (ZmaABF2). The resulting least standard deviation models were examined using PROCHECK [64] and Verfiy3D [65] services.

### 2.6. Docking of GH51 Proteins with Different Oligosaccharides

Structures of the xylan oligosaccharide (β(1,4)-d-xylopyranose-[α(1,3)-l-arabinofuranose]-di-β(1,4)-d-xylopyranoside (XAXX)) from the Tx-Abf-XAXX complex crystal structure (PDB ID: 2vrq) and free α-l-arabinofuranose co-crystallized with the MgGH51 structure were used for docking. The structures of the pectin arabinan fragment (tetra-α(1,5)-l-arabinofuranoside) and arabinogalactan II fragment (β(1,6)-d-galactopyranose-[α(1,3)-l-arabinofuranose]-di-β(1,6)-d-galactopyranoside) were constructed in the Glycam web tool (https://glycam.org/cb/) (accessed on 5 April 2022), then the puckering and the primary hydroxyl position in the terminal arabinofuranose was adjusted to that in the crystal structures. Molecular docking was performed in Autodock v4.2 software [66]. During docking, protein molecules remained rigid, while the linkages in the ligand were allowed to rotate sequentially. At the first step, the glycosidic linkages were kept flexible so that the terminal arabinosyl residue could adopt a position in the binding site and the xylosyl, galactosyl, or arabinosyl residues of the main chain could interact with the protein surface in close proximity. The best poses were used as initial geometries at the second step, where the hydroxyl groups of sugars were flexible to saturate hydrogen bonds with the protein. The docking poses with the most negative binding energy were selected for subsequent comparative analysis. Binding energies were calculated using Autodock as follows:(1)∆Gbinding=∆EvdW+∆EHbond+∆Eelec+∆Edesolv+∆Etors
where ΔEvdW is the contribution from van der Waals contacts, ΔEelec—from electrostatic interactions, ΔEHbond—hydrogen bonds, ΔEdesolv—desolvation, ΔEtors—entropic contribution.

The resulting plant structures in complexes with different ligands were analyzed and visualized in PyMOL software.

## 3. Results

### 3.1. The Identification of GH51 Genes in the Maize Genome

Eight maize genes (*ZmaABF1-8*) putatively encoding GH51 family enzymes were previously identified in the RefGenB73 AGPv3 genome assembly [14]. In the Zm-B73-REFERENCE-NAM-5.0 assembly [67] deposited in the GenBank, we identified five such genes and one pseudogene (*LOC109939272*) that matched *GRMZM2G393002* (*ZmaABF3*) (Table 1). The protein sequences encoded by *GRMZM2G120132* (*ZmaABF6*), *GRMZM2G452353* (*ZmaABF7*), and *GRMZM2G131440* (*ZmaABF8*) genes of the older genome version all displayed high similarity with that of *LOC109944448* of the newer assembly. However, the search of the GRMZM2G120132 protein in the GenBank revealed no information about the encoding gene, whereas both GRMZM2G452353 and GRMZM2G131440 corresponded to the *LOC109944448* gene. So, probably *GRMZM2G120132* (*ZmaABF6*) does not really exist, and *GRMZM2G452353* (*ZmaABF7*) and *GRMZM2G131440* (*ZmaABF8*) are the same gene designated as *LOC109944448* in the latest genome assembly. We kept the names given in the first study [14] to corresponding genes from the newer genome assembly (Table 1). The *LOC109944448* was named as *ZmaABF7* because the GRMZM2G452353 displayed a higher identity percent (99.4%) to translated *LOC109944448* compared to *GRMZM2G131440* (87.6%). However, *LOC109944448* encodes a short protein of 272 amino acids, and attempts to re-predict the gene were unsuccessful.

Four (*ZmaABF1*, *ZmaABF2*, *ZmaABF4*, and *ZmaABF5*) of five identified maize genes encoded the full-length proteins (Table 1). All of them had an Alpha-L-AF_C domain (IPR010720/PF06964) at the C-terminus and a glycoside hydrolase superfamily domain (Glycoside_hydrolase_SF, IPR017853) in the middle part of the sequence (Figure 1). Amino acid sequences ofZmaABF1, ZmaABF2, and ZmaABF4 proteins also contained a galactose-binding-like superfamily domain (Galactose-bd-like_sf, IPR008979) in the N-terminal region (Figure 1). Unlike the other sequences, ZmaABF2 did not have a signal peptide (Figure 1).

### 3.2. Abundance of GH51 Transcripts and Proteins in Maize Plants

In a multi-omics study by Walley and colleagues [51], transcripts of the *ZmaABF1* gene were found in almost all 23 maize tissues analyzed. They were particularly abundant in silks, seed embryos, and mature leaves (Figure 2).

The amount of the corresponding protein was high in anthers, seed pericarp and aleurone, in the juvenile and mature leaves, and in the root stele (Figure 2). The expression of the *ZmaABF2* gene was significantly lower in maize tissues compared with the *ZmaABF1* gene (Figure 2). However, transcript and protein levels were very high in the seed endosperm on days 8–10 after pollination (Figure 2, inset). The transcript level of the *ZmaABF4* gene was comparable to that of the *ZmaABF2* gene, but the corresponding protein product was detected only in silk and in the 5-day-old root stele at a very low level (Figure 2). The *ZmaABF5* gene also had low levels of mRNA in maize tissues, except for the seed pericarp and aleurone layer, where the corresponding protein was also detected at a significant level (Figure 2).

According to the expression data, *ZmaABF1* protein serves as the main isoform of α-l-arabinofuranosidases employed by maize in different tissues and at different developmental stages. *ZmaABF2* is required during the short period of endosperm development. *ZmaABF4* and *ZmaABF5* have moderate levels of transcripts across different tissues, but ZmaABF5 protein has been detected only in the aleurone and pericarp. The fine regulation of the set of expressed *α*-l-arabinofuranosidases in some tissues or during different development processes indicates possible differences in their functions. Such functions must be structurally and evolutionarily determined.

### 3.3. Phylogeny of GH51 Proteins of Various Plant Species

We searched for sequences encoding GH51 proteins in genomes and transcriptomes of different plants to assess their variability during evolution. All sequences found were checked on the proper length and presence of necessary domains and motifs (Appendix A). Such sequences are absent in the transcriptomes of red (*Rhodophyta*) and brown (*Phaeophyceae*) algae, and algae of *Chromista* and *Glaucophyta* phylum, deposited in the 1KP database. Thus, only the protein sequences of green plants were included in the resulting phylogenetic tree (Figure 3).

Sequences of GH51 are divided into three major clades: one of them includes *Chlorophyta* algae and the other two closely related clades include *Charophyta* algae and *Embryophyta* (land plants) representatives (Figure 3). The *Embryophyta* clade, in turn, is subdivided into several subclades, distinguishing hornworts, liverworts, mosses, ferns, lycophytes, gymnosperms, dicots, and monocotyledons (Figure 3). Most of the tree branches within the subclades are not very long, suggesting that the GH51 proteins of plants are highly conserved within the family.

The number of genes encoding GH51 proteins varies in different plant genomes. The genomes and transcriptomes of the *Chlorophyta* algae contain one or two genes encoding full-length GH51 proteins, while the *Charophyta* algae contain only one protein (Figure 3; Appendix A). Land plants possess one or two sequences, except for the transcriptome of *Thuja plicata* (Figure 3; Appendix A) and cereals, which have many isoforms. Basal monocotyledons (members of *Acorales*, *Alismatales*, and *Liliales* orders) possess only one or two genes encoding GH51 family members, whereas in other orders the number of isoforms has increased during evolution. The *Eleusine coracana* and *Oryza sativa* contain the greatest number of GH51 genes, nine and seven, respectively (Figure 4, Appendix A).

During evolution, the GH51 proteins of monocotyledons were divided into two large clades (Figure 3), both of which include sequences of both commelinids and non-commelinids. Commelinids include *Arecales*, *Zingiberales*, *Commelinales,* and *Poales* orders and are distinguished by the special composition of the primary cell wall and the presence of ferulic acids [69,70].

The clade II includes sequences with a substitution of acid/base glutamic acid (E) (underlined) for glutamine (Q) in the conserved G-N-E motif [71], while in clade I this substitution is absent (Figure 4). However, the *Zostera marina* (eelgrass) enzyme joined clade II despite the absence of acid/base substitution, and one enzyme of *Spirodela polyrhiza* (duckweed) with a substitution joined clade I (Figure 4). The phylogenetic analysis of a broader list of monocotyledons confirmed the division of *Liliopsida* GH51 proteins into clades I and II (Figure 4). Clade II includes enzymes of the *Poales*, *Asparagales*, and *Zingiberales* orders, and in almost all of them the catalytic acid/base E is replaced by Q.

### 3.4. N-Terminal Domain of GH51 α-l-arabinofuranosidases

Three out of four full-length maize GH51 sequences contained a domain at the N-terminus that was annotated as galactose-binding-like superfamily domain (Figure 1). Many other plant sequences used for phylogenetic analysis were also predicted to contain this domain or a carbohydrate binding module of the family 4 (CBM4) domain (PF02018) (Figure 3; Appendix A). A homologous N-terminal domain was found in the MgGH51 enzyme of the fungus *M. giganteus*. It forms a ten-strand β-sandwich structure similar to the bacterial CBM4 domains [31] (Figure 5A,B, highlighted in purple). However, unlike the bacterial CBM4s that bind cellulose, laminarin, and xylan oligosaccharides [72,73,74,75], the N-terminal domain of MgGH51 was unable to bind saccharides and was named a CBM4-like domain [31]. We used this name in our study.

The N-terminal sequences of ZmaABF1 and AtARAF1, which are homologous to the CBM4-like domain of MgGH51, were used for BLASTP search in the GenBank database, excluding *Viridiplantae* (green plants). The search resulted in bacterial, fungal, and six insect (*Hexopoda*) sequences all belonging to the GH51 family indicating the specificity of the CBM4-like domain to that glycoside hydrolase family. The insect sequences have 30% identity with plant, 35–40% identity with fungal, and 41–48% identity with bacterial enzymes, so horizontal gene transfer is most likely.

We created the CBM4-like domain profile in HMMer-3.3 using multiple alignment of 16 fungal, 11 bacterial, and 111 plant sequences possessing the N-terminal domain sequence. The beginning and the end of the CBM4-like domain were established by L-x-V-D and S-x-x-P conserved motifs, respectively (Appendix A). The S-x-x-P motif is located on the loop separating the CBM4-like domain from the others in the MgGH51 structure [31]. The generated CBM4-like domain profile was represented in 119 of 133 (89.5%) GH51 plant sequences (Appendix A) that were used for phylogenetic analysis earlier (Figure 3).

To estimate the proportion of GH51 sequences that contain a CBM4-like domain in bacteria, fungi, and insects, we downloaded all available protein sequences of these organisms that were longer than 200 residues and contained either the PF06964 domain or both the PF02018 and PF06964 domains from the Uniprot database. Among the 13584 bacterial and 2189 fungal GH51 protein sequences, 3362 (24.7%) and 1354 (61.8%), respectively, possessed a generated CBM4-like domain profile. All insect sequences (6 out of 6, 100%) contained a CBM4-like domain. A total of 866 of 1027 (84.3%) and 402 of 437 (91.9%) plant sequences with the PF06964 domain deposited in the Uniprot and Phytozome databases, respectively, possessed the CBM4-like domain. The superposition of the MgGH51 arabinofuranosidase carrying a CBM4-like domain and bacterial GH51 arabinofuranosidase (Tx-Abf) lacking this domain is shown in Figure 5A.

Phylogenetic analysis of the 138 CBM4-like domain sequences used to create the domain profile showed that the sequences of fungi and land plants are the most distant from each other (Appendix A). The bacterial N-terminal domains alternate with those of the algae *Chlorophyta* and, probably, *Chlorophyta* acquired it during gene horizontal transfer. The N-terminal domain of the algae *Charophyta* occupies the same clade as *Lycophyte* (Appendix A).

To establish the differences between bacterial CBM4s and CBM4-like domains, we aligned their protein sequences. The sequences of bacterial CBMs of family 4 with resolved crystal structures [72,73,74,75] were compared with those of CBM4-like domains of some bacteria, plants, and fungi (Appendix A). None of the amino acid residues which were reported as crucial for recognition and accommodation of carbohydrate substrates in bacterial CBM4s ([72,73,74,75]; Appendix A, green background) were conserved in CBM4-like sequences tested (Appendix A). In contrast, the G-x-x-F-E-x-I-x-x-x-G-x-G-G motif which is conserved in CBM4-like domains (Appendix A, black frame) is absent or significantly altered in bacterial CBM4s (Appendix A, black frame). This motif is located in the β-strand that permeates the entire spatial structure of the MgGH51 enzyme and forms the bottom of its catalytic pocket (Figure 5A,B, highlighted in blue) [31].

The polar amino acid residues of bacterial CBMs of family 4 interact with the OH groups of sugars, and the aromatic radicals of tryptophan (W), tyrosine (Y), and phenylalanine (F) residues with the sugar rings through a stacking-like interaction [76]. These amino acids form a cleft-like structure in which oligosaccharides are placed, such as in *Rhodotermus marinus* xylanase (RmCBM4-2) [73] (Figure 5C). In the MgGH51 and ZmaABFs, most of the residues homologous to these substrate-binding amino acids of CBM4-like domain are aliphatic (Appendix A, Figure 5D). Moreover, the arginine (R93) of the MgCBM4-like surface is located on a loop next to the putative cleft-like pocket and crosses it, making the MgCBM4-like surface rather flat (Figure 5E) in contrast to resolved bacterial CBM4s (Figure 5C).

The CBM4-like domain of ZmaABF1 compared to the fungal CBM4-like demonstrates a cleft-like surface similar to that of RmCBM4-2 (Figure 5F); however, it is narrower than that in RmCBM4-2 and, probably, cannot accommodate the sugar. In the CBM4-like domains of ZmaABF2 and ZmaABF5, the cleft-like surface is almost absent as in the CBM4-like of MgGH51. The putative cleft is populated by residues with different functional groups (Figure 5F). Thus, according to alignments and modeling, neither the CBM4-like domain of MgGH51 nor the CBM4-like domains of maize GH51 proteins are capable of binding oligosaccharides in a manner similar to that displayed by bacterial CBM4s.

### 3.5. Three-Dimensional Models of GH51 Proteins and Molecular Docking

Three-dimensional models of ZmaABF1, ZmaABF2, and ZmaABF5, and additionally AtARAF1, were constructed using the crystal structure of the fungus MgGH51 as a template. ZmaABF4 was discarded because of the lack of significant accumulation of transcripts and corresponding proteins in maize tissues (Figure 2; [51]). The sequence identity between maize proteins, AtARAF1, and MgGH51 ranged from 32.5 (ZmaABF2) to 34.3% (ZmaABF5), which is sufficient for homologous modeling [77]. All models of maize proteins, AtARAF1 of *A. thaliana*, structures of fungal MgGH51, and bacterial Tx-Abf with replacement of catalytic acid/base E with Q and wild type enzyme were subjected to molecular docking with a number of substrates (Table 2). In silico prediction of the specificity of glycoside hydrolases to a particular substrate is effective [78], as has already been demonstrated, including for the α-l-arabinofuranosidases of GH51 [79]. However, the results of such modeling should be treated with caution, since they are derived from predicted protein structures. During homology modeling, the orientation of the arabinosyl residue in the active site was preserved as in the X-ray structure of the MgGH51-arabinose complex. Thus, a single permissible binding mode of the terminal arabinose was observed in all proteins; therefore, the difference in binding energies is mainly due to the interaction of the protein with the oligosaccharide backbone.

Among the studied proteins, bacterial enzyme Tx-Abf displayed the highest affinity to the fragment of arabinan (Table 2). Both Tx-Abf (with E/Q mutation and wild type enzyme) and fungal MgGH51 also exhibited high affinity to the monosubstituted fragment of xylan; however, judging from slightly lower ΔG absolute value, arabinan was a less favorable ligand for MgGH51 than for Tx-Abf (Table 2). The arabinogalactan II fragment was the least favorable substrate for Tx-Abf among those tested (Table 2). The Tx-Abf is the xylan-utilizing enzyme as it was also shown biochemically [70].

Maize proteins had lower binding energies when interacting with monosubstituted xylan when compared with bacterial or fungal enzymes. Their interaction with arabinan was even less favorable (Table 2). Surprisingly, the model of AtARAF1 also displayed higher affinity to monosubstituted xylan than that to its known substrate [34], arabinan (Table 2). All attempts to simulate the interaction of plant GH51 models with disubstituted xylan fragments were unsuccessful.

In maize models, the conservative catalytic E342 and E419 formed the hydrogen bonds to the arabinose moiety of XAXX (Q349 and E426 in ZmaABF5) (Figure 6A,B). The E23, N341 (N348 in ZmaABF5), and Y392 (Y399 in ZmaABF5) also formed hydrogen bonds to arabinose and xylose (Figure 6B). The same amino acids were involved in the binding of arabinan fragments in maize models; however, in ZmaABF1, this substrate was additionally bound by H288 and E226 residues (Figure 6C) and in ZmaABF5 by N230 (data not shown). Probably, these additional hydrogen bonds from H288 and E226 provide a higher affinity of ZmaABF1 to arabinan compared to other maize GH51 models (Table 2).

The structure of the catalytic (β/α)_8_-barrel domain in maize models is almost the same as that of MgGH51, except for ZmaABF2. The ZmaABF2 protein sequence lacks a fragment of 19 residues which forms a β-strand in the catalytic (β/α)_8_-barrel domain of ZmaABF1, ZmaABF5, and MgGH51 (Figure 6D). In MgGH51, this β-strand is involved in the formation of a flat surface of a xylan-binding catalytic groove with a deep pocket for arabinose (Figure 5B). This surface is organized similarly in ZmaABF1 and ZmaABF5 (Figure 6A). The absence of the 19-residue fragment makes that surface of ZmaABF2 shallow and more heterogeneous than that in MgGH51 (Figure 6A). The putative arabinose-binding pocket in ZmaABF2 is looser and not as deep as in ZmaABF1 and ZmaABF5 (Figure 6A). Nevertheless, according to molecular docking results, both the narrow catalytic grooves of ZmaABF1 and ZmaABF5 and the wide groove of ZmaABF2 are unable to accept the xylooligosaccharide di-substituted with arabinose (data not shown).

## 4. Discussion

The five maize genes encoding GH51 proteins were identified in the latest version of the maize genome (Table 1) compared to the eight identified previously [14]. The gene identification results depend on the genome assembly quality. The identification and analysis of maize GH51 genes are useful for further studies of these plant enzymes, which may include the construction of recombinant proteins and other techniques of molecular biology.

During early plant evolution, α-l-arabinofuranosidases of the GH51 family probably appeared in the genomes of the green algae *Chlorophyta*, since we did not find genes for these proteins in red, brown algae, and other basal eukaryotes in used databases. However, the GH51 proteins are also presented in fungi and some insects. Despite the absence of GH51 proteins, arabinose is present in the cell walls of brown algae, for example, as part of AGPs arabinogalactans [80], however, arabinose is a minor monosaccharide in the cell walls of red algae compared to brown and green algae [81]. In contrast, in *Chlorophyta* algae, arabinose is the major terminal-linked monosaccharide of carbohydrates [82]. Polymers that form the shell structure of the green alga *Botryococcus* have arabinose and galactose in significant quantities [83]. Arabinose is a part of arabinomannan [84], a unique type of arabinogalactan [85], and glucuronan with arabinose-containing side chains [86] and N-glycosylation chains in *Chlorella* [87]. The increase in the amount of arabinose in various polymers in *Chlorophyta* could form the demand for new α-l-arabinofuranosidases and result in the emergence of the GH51 family. However, the apparent absence of corresponding genes in other algae and basal eukaryotes may be the result of scarce genomic data for them compared to *Viridiplantae.*

Since their appearance in *Chlorophyta*, GH51 α-l-arabinofuranosidases have moderately changed during plant evolution (Figure 3). A few whole-genome duplication events that occurred in monocotyledonous plants of the order *Poales* [88] probably lead to an increase in the number of GH51 genes in cereals (Figure 3 and Figure 4; Appendix A). For example, barley has five genes encoding GH51 proteins, sorghum has six, and rice has seven (Figure 3 and Figure 4; Appendix A). The increased number of putative α-l-arabinofuranosidases in cereals, along with a large number of α-l-arabinosyl- and β-d-xylosyltransferases [89,90], is probably necessary for the synthesis and modification of arabinoxylans with a complex side chain pattern [91]. The idea of the special development of GH51 in species possessing complex xylans in cell walls is supported by the fact that the number of genes encoding GH51 is also increased in the majority of analyzed representatives of the order *Asparagales* (Figure 4). The *Asparagales* are non-commelinid, i.e., their primary cell walls are of the so-called type I and are abundant in pectins and xyloglucans, but not in arabinoxylans and mixed-linkage glucans characteristic of type II cell walls common to commelinids [69]. Despite this, *Asparagales* have cell wall xylans containing a terminal 1,2-linked arabinopyranoside as a side substituent in addition to the typical glucuronic acid [92]. Thus, the increased amount of α-l-arabinofuranosidases in *Asparagales* may also be directed toward cell wall xylan modification.

A progressive step in GH51 protein evolution seems to have been the division of monocot proteins into two large clades I and II (Figure 3 and Figure 4). Clade II includes sequences of only three orders (*Asparagales*, *Zingiberales*, and *Poales*) of monocotyledons (Figure 4). The absence of clade II sequences in genomes of other monocot orders indicates that this clade probably could have arisen during the whole-genome duplication of the *Asparagales*, *Zingiberales*, and *Poales* which took place approx. 66 million years ago [88]. These orders separated from each other about 120 million years ago [93]. This means that their molecular evolution had long been independent by the time of the genome duplication 66 million years ago. In this context, it is particularly interesting that most of the representatives of clade II carry the catalytic acid substitution (Figure 4). A similar replacement of acid/base glutamic acid with glutamine in Tx-Abf resulted in a more than 100-fold decrease in specific activity compared to native enzyme in vitro [71]. According to our docking study, this decrease in activity should not be due to a decrease in affinity to the substrate, since the indicated replacement had almost no effect on it (Table 2). Some enzymes, such as the GH1 myrosinases or GH92 α-mannosidases, can function efficiently in the absence of an acid/base catalyst [94,95], probably by using the substrate’s own acid catalyst [96]. The possibility of using a similar mechanism by GH51 enzymes from clade II monocotyledons should be studied separately. Another possibility is that the catalytic site of clade II GH51 is indeed disrupted, which has already been reported for some glycoside hydrolases [97]. The significance of acid/base substitution in plant GH51 proteins requires further studies.

The enzymes of clade I seem to be more important and more active compared to clade II members. Barley HvAXAH1-2 enzymes belonging to clade I (Figure 3) showed high catalytic rates in vitro, and the corresponding genes were expressed at high levels in many tissues [68], as were maize GH51 genes (*ZmaABF1*; *ZmaABF2*) from clade I (Figure 2 and Figure 3). The ZmaABF1 protein and corresponding transcripts were found in almost all tissues analyzed by Walley and colleagues [51]. Protein accumulation was high in mature leaves, seed pericarp, and the root stele, but not in the meristem and elongation growth zones of root (Figure 2) [51]. This is consistent with the recently described co-expression of *ZmaABF1* with the genes of secondary cell wall cellulose synthases in maize root [35], since mature leaves, seed pericarp, and root stele also develop secondary cell walls. The modeled maize ZmaABF1 displayed the highest affinity to both arabinoxylan and arabinan fragments among other maize enzymes studied, comparable to estimates for MgGH51 (Table 2), suggesting that ZmaABF1 is efficiently adapted to handle both these substrates. At the same time, the catalytic site of ZmaABF1 does not seem to be able to accommodate the arabinose of disubstitution of the arabinoxylan backbone. Thus, ZmaABF1 appears to serve as the main isoform of GH51 α-l-arabinofuranosidases employed by maize in different tissues, especially those developing secondary cell walls. Monosubstituted xylan moieties probably serve the main target for this enzyme; however, its affinity to α-l(1,5)-arabinan is also significant.

ZmaABF2 appears to be a seed-specific enzyme. The *ZmaABF2* gene is expressed at low levels in most maize tissues [51,52], with the exception of seeds at stage 6–10 DAP (Figure 2) [52], where a peak of expression was found in the endosperm at 6 DAP [98]. ZmaABF2 protein accumulation also peaked only in the endosperm at 8–10 DAP (Figure 2) [51]. During 3–12 DAP, the maize endosperm is cellularized, suggesting intense cell division and formation of the aleurone layer (6–10 DAP) [99]. Cellularization is accompanied by the deposition of cell wall components as observed in barley seeds at the 6 DAP stage [100]. The modeled ZmaABF2 exhibited higher affinity for fragments of arabinoxylan and arabinogalactan than that for arabinan (Table 2), indicating that the latter may be the least favorable substrate. The absence of a predicted signal peptide in ZmaABF2 (Figure 1) may indicate that this enzyme accumulates first within some seed cells separately from its primary substrates and then acts on them only at the stage of seed germination. ZmaABF2 differs from other maize GH51 proteins by the absence of a 19-amino acid fragment in the protein sequence (Appendix A), which led to the formation of an additional hole near the catalytic pocket; however, it did not affect the catalytic pocket itself (Figure 6D). Nevertheless, ZmaABF2 with changed topology of surface is unable to bind the di-substituted arabinose fragment of xylan in silico (data not shown). This deletion of 19 residues is not present in ZmaABF2 protein sequences in earlier assemblies of the maize genome, so it is difficult to characterize this deletion as unique to ZmaABF2 among other maize enzymes. In general, among GH51 α-l-arabinofuranosidases, only a few, mostly fungal, can detach arabinose from di-substituted xylans [101]. Plant enzymes, if they are capable of mediating this reaction, catalyze it at a low rate [21] and, apparently, these enzymes are absent among maize GH51 proteins.

The barley clade II members, HvAXAH3-5 (Figure 3), showed negligible in vitro catalytic rates against different arabinose-containing polysaccharides and low expression levels of the corresponding genes [68]. Transcription of maize clade II members (*ZmaABF4* and *ZmaABF5* genes) was also low, and accumulation of the corresponding protein was negligible, except for significant accumulation of the ZmaABF5 protein in the seed pericarp (Figure 2; [51]). The acid/base mutation did not occur long ago by evolutionary standards, but it has been preserved in the genomes of various monocotyledons for millions of years. The GH51 proteins of clade II may have minor functions in cereals or require specific conditions for catalysis, or they may play a role as reserve enzymes in cereals. Another possibility is that, having lost their hydrolytic function, these proteins have acquired some other function [97,102]. Anyway, the physiological significance of the mutated GH51 proteins requires additional studies.

The third N-terminal CBM4-like domain was found only in sequences of plant, fungal, and bacterial GH51 proteins and it is probably unique to the enzymes of this family. This domain is present in 84–92% (depending on the search database) of analyzed plant GH51 sequences (Appendix A), but only in 62% and 25% of fungal and bacterial sequences, respectively. The CBM4-like domain of plant GH51 family proteins appeared in green (*Chlorophyta*) ancient algae during evolution and passed to *Charophyta* algae and land plants (Figure 3; Appendix A). The sequences of CBM4-like domains of plants and fungi are the most remote from each other according to phylogenetic analysis results (Appendix A). The bacterial CBM4-like domains occupy an intermediate position between fungi and plants; however, the domain of analyzed *Chlorophyta* algae is closely related to some bacteria (Appendix A).

The evolution of CBM4-like domains and whole GH51 sequences in land plants is generally similar. CBM4-like domains of angiosperms are distant from gymnosperms, which, in turn, are distant from ferns and parts of *Lycophyte* with significant values of ultrafast bootstrap support (Appendix A), as in the case of whole sequences (Figure 3). CBM4-like domains of liverworts and hornworts are more basal (Appendix A), although in the case of whole GH51 sequences, mosses share this position with them (Figure 3). CBM4-like domains of mosses, parts of *Lycophyte*, and the alga *Charophyta* are interspersed with each other with low branch support values, indicating minor differences in the CBM4-like domain sequences in these groups (Appendix A). CBM4-like domains of *Charophyta* algae have evolved considerably compared to *Chlorophyta*. This coincides with the theory suggesting the *Charophyta* are the closest ancestors of land plants [44,103]. The topologies of phylogenetic trees constructed separately for CBM4-like domains (Appendix A) and for full-length GH51 proteins of monocotyledons (Figure 3) are similar (data not shown). However, the clade II is more basal compared to clade I (data not shown) in tree for CBM4-like domains (Appendix A) in contrast to whole-sequence phylogenetic analysis (Figure 4).

The various CBMs in the glycoside hydrolases facilitate substrate recognition and increase affinity, especially for insoluble surfaces such as cellulose, in microbial enzymes. The replacement of native CBMs to another one is able to change the substrate specificity of glycoside hydrolase [104]. The CBMs of family 4 belong to the type B of CBMs, which recognize the glycan chains up to six sugar moieties. The binding cleft of CBMs of this type can be either deep or shallow [76]. The aromatic and charged amino acid residues are responsible for ligand binding and orientation [76]. The surface of the CBM4-like domain in the α-l-arabinofuranosidase of MgGH51 and maize GH51 proteins is shallow and devoid of necessary residues (Figure 5C–F). The CBM4-like domain in these proteins is probably incapable of binding saccharides. However, this domain was retained in the GH51 sequences of plants and evolved with them. Thus, the presence of the CBM4-like domain is probably important for plant GH51 proteins, but the understanding of its function requires additional biochemical studies.

## Figures and Tables

**Figure 1 life-13-00266-f001:**
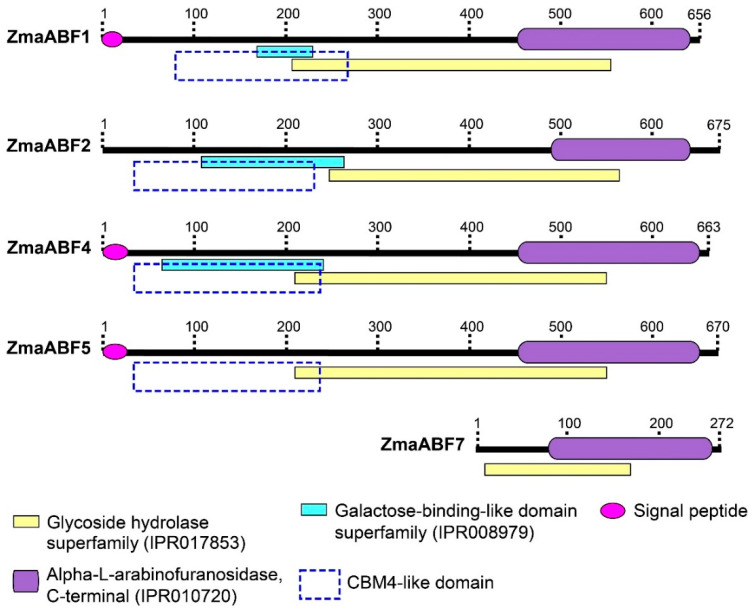
Domain organization of GH51 proteins encoded in Zm-B73-REFERENCE-NAM-5.0 maize genome assembly. The numbers above the domain schemes indicate the number of amino acid residues.

**Figure 2 life-13-00266-f002:**
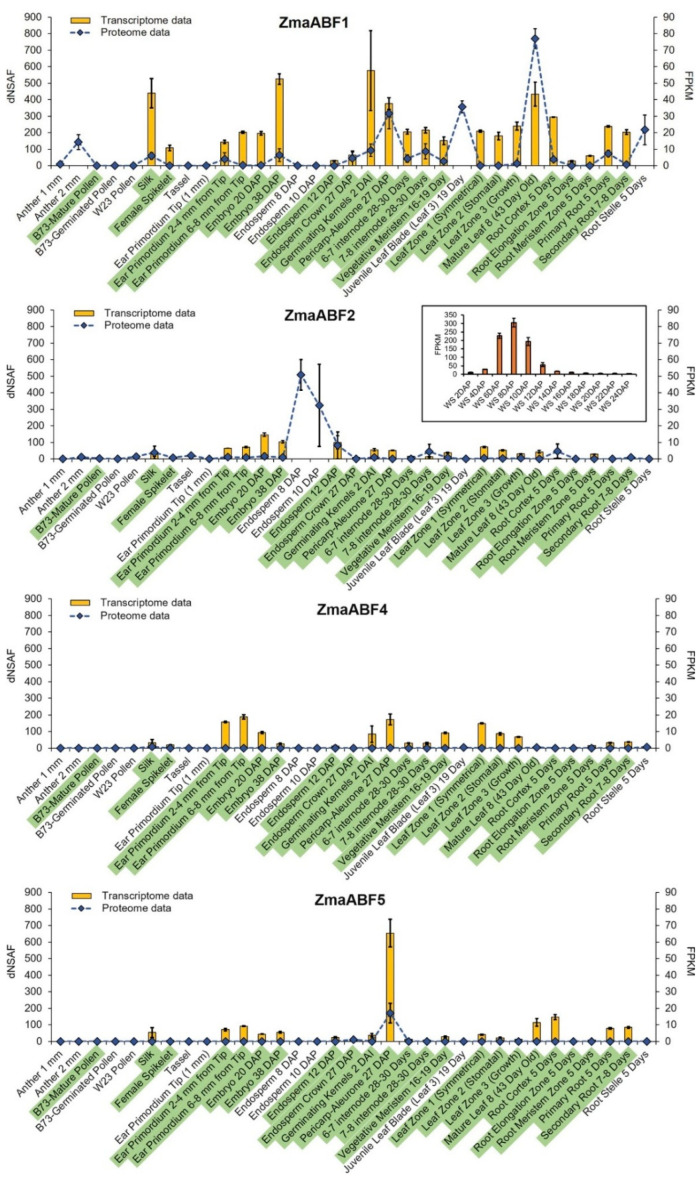
Expression levels of α-l-arabinofuranosidase genes of the GH51 family in various maize tissues according to Walley et al. [51] and Hoopes et al. [52]. Transcriptomic data are presented as FPKM (fragments per kilobase of transcript per million fragments mapped) values, proteomic data are presented as dNSAF (distributed normalized spectral abundance factor) values, both with standard deviations. The green background for some sample names marks those for which Walley and colleagues [51] obtained both transcriptomic and proteomic data, whereas for the remaining samples only proteomic data were obtained. WS, whole seed; DAP, days after pollination; DAI, days after imbibition.

**Figure 3 life-13-00266-f003:**
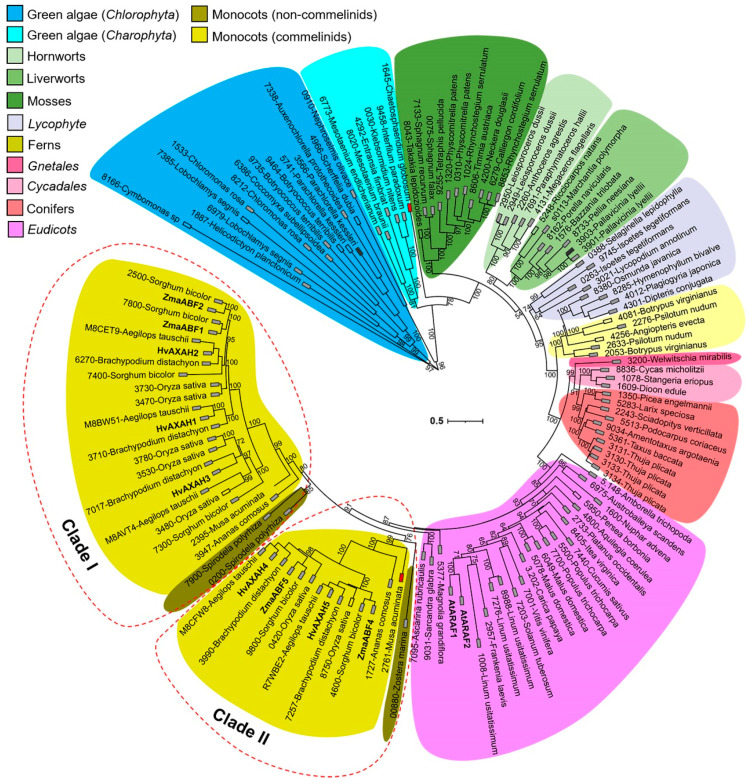
Maximum likelihood phylogenetic tree of plant full-length protein sequences of the GH51 family in circular representation. Characterized plant enzymes are highlighted in bold [14,34,68]. Red dotted lines surround two clades of monocotyledons. Colored rectangles next to the protein names show the presence of a third N-terminal CBM4-like domain in addition to α-l-arabinofuranosidase (Alpha-L-AF_C (PF06964)) and Glycoside_hydrolase_SF (IPR017853) domains. Among them, the gray rectangles show proteins with a CBM4-like domain. Black and red rectangles show sequences with CBM4-like domain also predicted in the InterProScan as galactose-bd-like_sf or CBM_4_9 (PF02018) domains, respectively. Blank rectangles indicate the absence of N-terminal CBM4-like domain. The architecture of the domains of each sequence as well as the full names of the proteins are given in Appendix A. Numbers indicate the ultrafast bootstrap support values.

**Figure 4 life-13-00266-f004:**
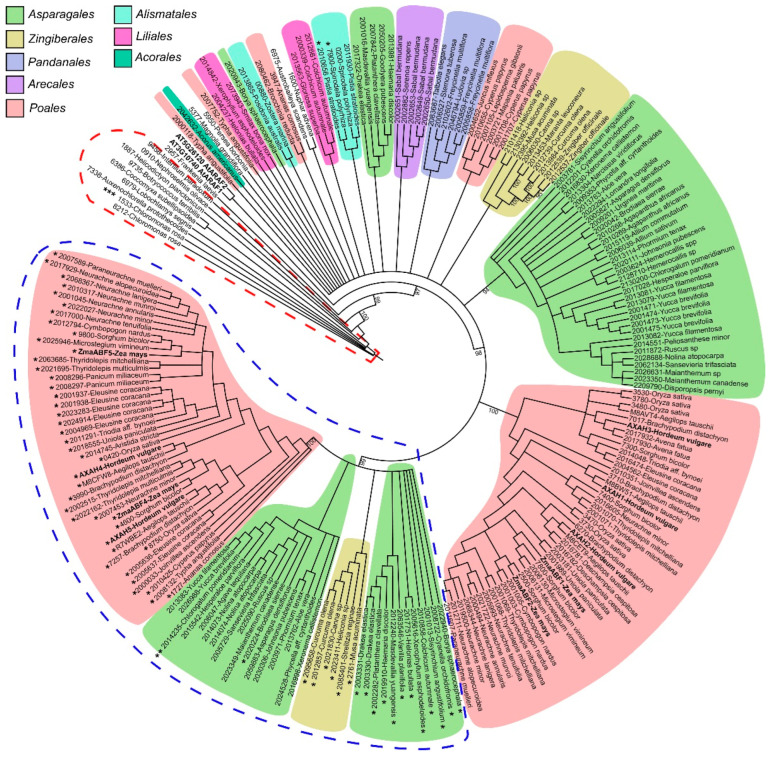
Maximum likelihood phylogenetic tree of GH51 full-length proteins of monocotyledons (*Liliopsida*). Figures of different color denote representatives of different orders of monocotyledons. The red dotted line indicates sequences of *Chlorophyta* algae, which were used as an outgroup along with *AtARAF1-2* of *Arabidopsis thaliana*, some basal angiosperms (*Persea borbonia*, *Magnolia grandiflora*, *Frankenia laevis*), and basal eudicots (*Nuphar advena*, *Austrobaileya scandens*). The clade II of GH51 proteins is shown by the blue dotted line. The sequences of other monocotyledonous orders belong to clade I. The asterisks indicate the sequences with the G-N-Q motif instead of G-N-E. The double asterisk indicates the sequence with G-N-D and triple asterisk with G-S-N motifs instead of G-N-E. The presence of the N-terminal CBM4-like domain in sequences is given in Appendix A. Branches with ultrafast bootstrap support values less than 90 were removed.

**Figure 5 life-13-00266-f005:**
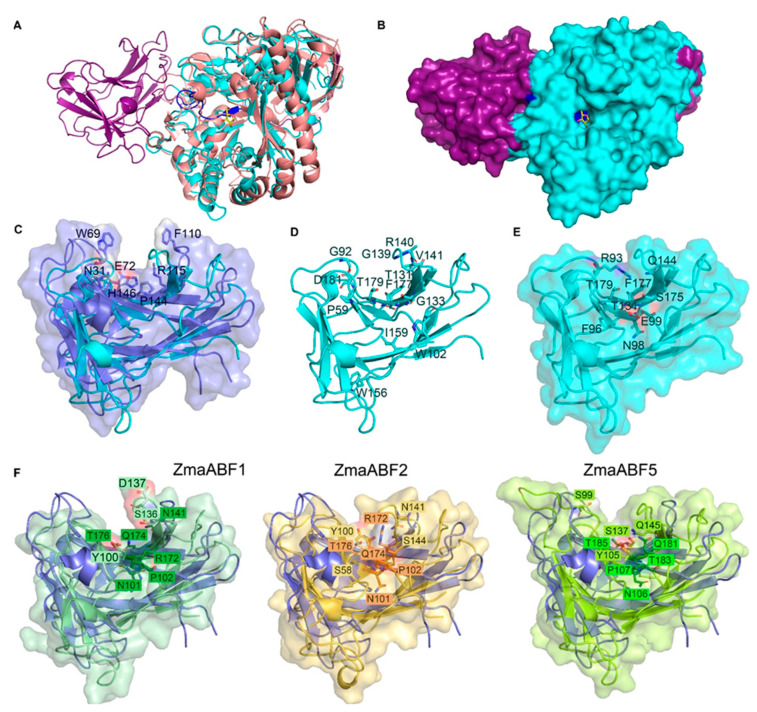
Spatial structures of bacterial and fungal arabinofuranosidases of GH51 family (**A**,**B**); bacterial CBM4 and fungal and plant CBM4-like domains (**C**–**F**). (**A**) The cartoon representation of superimposition of bacterial GH51 Tx-Abf (PDB ID: 2vrq, salmon) from *Thermobacillus xylanilyticus* and fungal MgGH51 (PDB ID: 6pzy, cyan) from *Meripillus giganteus*. The CBM4-like domain of MgGH51 (MgCBM4-like) is highlighted in purple. The conserved G-x-x-F-E-x-I-x-x-x-G-x-G-G motif is highlighted in blue. Arabinose co-crystallized with the MgGH51 structure is represented as sticks with yellow C atoms and red O atoms. (**B**) MgGH51 in the surface representation and arabinose in sticks representation colored in the same way as in (**A**). (**C**) Superimposed CBM4-2 from *Rhodotermus marinus* xylanase (RmCBM4-2; PDB ID: 1k42; dark purple) and CBM4-like domain from MgGH51 (blue). CBM4-2 from *R. marinus* is shown as a cartoon and surface. RmCBM4-2 amino acids involved in xylooligosaccharide recognition are shown as purple sticks with red O atoms, blue N atoms, and purple C atoms. (**D**) MgCBM4-like domain in cartoon representation. Amino acid residues of the MgCBM4-like domain homologous to the substrate-binding residues of bacterial CBMs of family 4 are shown as sticks with red O atoms, blue N atoms, and cyan C atoms. (**E**) MgCBM4-like domain in cartoon and surface representations. Amino acids with functional groups in close proximity to the putative substrate-binding cleft are shown as sticks with red O atoms, blue N atoms, and cyan C atoms. (**F**) Superimposed structures of RmCBM4-2 (cartoon representation; dark purple) and the CBM4-like domains of maize GH51 proteins (cartoon and surface representations of dark green, yellow-orange and lime colors for ZmaABF1, ZmaABF2, and ZmaABF5, respectively). Amino acid residues of maize CBM4-like domains homologous to (lighter labels) and those occupying the same spatial positions (darker labels) as the substrate-binding residues of bacterial CBMs of family 4 are shown as sticks.

**Figure 6 life-13-00266-f006:**
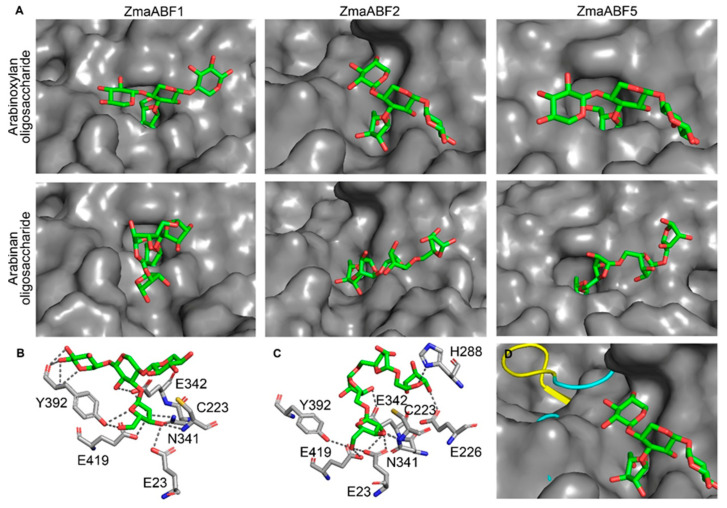
Three-dimensional models of ZmaABF1, ZmaABF2, and ZmaABF5 of the GH51 family and molecular docking results. (**A**) The spatial models of ZmaABF1, ZmaABF2, and ZmaABF5 in surface representation with the docked β(1,4)-d-xylopyranose-[α(1,3)-l-arabinofuranose]-di-β(1,4)-d-xylopyranoside or tetra-α(1,3)-l-arabinofuranoside, shown as sticks, where C atoms are green and O atoms are red. (**B**) The amino acid residues of ZmaABF1 that form the hydrogen bonds with arabinoxylan and arabinan oligosaccharides are shown as sticks with red O atoms, blue N atoms, yellow S atoms. C atoms are gray in amino acids and green in sugars. The bonds are shown by gray dotted lines. (**C**,**D**) The superimposition of ZmaABF2 (in surface mode) on the ZmaABF1 (in cartoon mode), where 19-residue insertion is highlighted in yellow. The docked arabinoxylan oligosaccharide is shown as sticks, where C atoms are green and O atoms are red.

**Table 1 life-13-00266-t001:** The maize genes encoding proteins of GH51 family. aa, the number of amino acids of corresponding proteins; nd, no data.

RefGenB73 AGPv3		Zm-B73-REFERENCE-NAM-5.0			
Gene Locus IDs	Gene Name Used in the Study by Kozlova et al. (2015) [14]	Gene Locus/ Protein IDs	Gene Names Used in Present Study	Chromosome Number	The Protein Length, aa
*GRMZM2G006130*	*ZmaABF1*	*LOC100193890*/ NP_001337023	*ZmaABF1*	3	656
*GRMZM2G077299*	*ZmaABF2*	*LOC100282597*/ XP_035822820	*ZmaABF2*	4	675
*GRMZM2G393002*	*ZmaABF3*	*LOC109939272* (pseudogene)	nd	3	nd
*GRMZM2G097277*	*ZmaABF4*	*LOC100193089*/ NP_001345560	*ZmaABF4*	2	663
*GRMZM2G180889*	*ZmaABF5*	*LOC103634588*/ XP_035819499	*ZmaABF5*	1	670
*GRMZM2G120132*	*ZmaABF6*	nd	nd	nd	nd
*GRMZM2G452353*	*ZmaABF7*	*LOC109944448*/ XP_020404813	*ZmaABF7*	2	272
*GRMZM2G131440*	*ZmaABF8*				

**Table 2 life-13-00266-t002:** The molecular docking results for plant, bacterial, and fungal GH51 enzymes. The wild type of Tx-Abf is marked by asterisk. Xyl*p,* xylopyranose; Ara*f*, arabinofuranose; Gal*p*, galactopyranose; ND, not determined.

	The Docking Score ΔG (kcal mol^−1^)		
	β(1,4)-d-Xyl*p*-[α(1,3)-l-Ara*f*]-β(1,4)-di-d-Xyl*p*	tetra-α(1,5)-l-Ara*f*	β(1,6)-d-Gal*p*-[α(1,3)-l-Ara*f*]-β(1,6)-d-di-Gal*p*
Tx-Abf (2vrq)	−8.46	−9.23	−6.25
Tx-Abf ***** (2vrk)	−8.11	ND	ND
MgGH51 (6zpy)	−7.18	−4.96	ND
ZmaABF1	−6.57	−4.52	ND
ZmaABF2	−5.52	−1.83	−3.50
ZmaABF5	−6.21	−3.82	ND
AtARAF1	−7.00	−2.69	ND

## Data Availability

Not applicable.

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
