# Peer review of "The In Silico Characterization of Monocotyledonous α-l-Arabinofuranosidases on the Example of Maize"

_life, 2023, doi:10.3390/life13020266_

Round 1
Reviewer 1 Report
Glycoside hydrolases are widespread and well-studied enzymes, grouped into 173 families based on homology (CAZy database). The MS authors investigated GH51 family proteins from maize and compared them with their orthologues from other plants. Phylogenetic analysis of GH51 and CBM4 protein domains has been performed, as well as the homology modeling. Evolution and functional roles of GH51 family proteins are discussed.
– I think the title is not appropriate. It does not fully reflect the content of the article. Most probably two out of four analyzed maize proteins do not have any enzymatic activity. I suggest to formulate the title in more general: mention monocotyledons but not only maize.
– Lines 16-17. I suggest to remove this sentence (The appearance of GH51 α-L-arabinofuranosidases in plants coincides with the appearance of arabinose in their cell walls.). It is too speculative.
– All described enzymatic activities for the GH51 family (see CAZy database) should be listed in the Introduction. Authors should explain somewhere why they consider maize proteins as α-L-arabinofuranosidases.
– Line 33. The family list is incomplete (see CAZy database).
– Lines 128-132 and 302-305. I suggest to repeat the BLASTP search for CBM4-like domains and provide the current statistics for different taxonomic groups (bacteria, fungi, plants, insects).
– I suggest to move the 3.1. section to the Supplementary Materials. Add data about chromosome number in Table 1. Please explain if ZmaABF3 and ZmaABF6 are encoded in the maize genome or not. Do you think that ZmaABF7-8 is a pseudogene? Does it have orthologues in other plants?
– I think the Figure 1 is not appropriate and should be removed. Do not try to compare different genome assemblies by phylogenetic analysis. Use it only for reconstructing evolutionary events.
– I advise to expand the 3.3. and 3.4. sections and to reduce all others. Move Figure S1 to the body of the article.
– I suggest to move the Figure 4 to the Supplementary Materials.
– Try to compare topology of GH51 and CBM4 phylogenetic trees.
Reviewer 2 Report
line 34: pol- => poly-
line 54: as was shown => as it was shown
line 89: a-L-arabinofuranosidase => a-L-arabinofuranosidase
line 132: ver- => veri-
line 158: a-L- => a-L-
line 162: a-L- => a-L-
line 193: fam- =>: fami-
line 223: gray.backgrounds => gray backgrounds
line 524: arab- => arabi-
Round 2
Reviewer 1 Report
The manuscript has been substantially revised. Most of my comments have been taken into account. However, two of my comments are partially ignored.
====
Point 5: Lines 128-132 and 302-305. I suggest to repeat the BLASTP search for CBM4-like domains and provide the current statistics for different taxonomic groups (bacteria, fungi, plants, insects).
Response 5: We have updated some statistics, but you should keep in mind that it will become irrelevant in a few days because of the high rate of depositing of new sequences. That is why we prefer to specify percentages rather than absolute values.
I did not find in the "Materials and Methods" section that any database were accessed in 2022. There are also no statistics on insects in the manuscript.
====
Point 6: I suggest to move the 3.1. section to the Supplementary Materials. Add data about chromosome number in Table 1. Please explain if ZmaABF3 and ZmaABF6 are encoded in the maize genome or not. Do you think that ZmaABF7-8 is a pseudogene? Does it have orthologues in other plants?
Response 6: We have shortened that section considerably. Table 1 has been significantly edited. Chromosome numbers have been added. The genes ZmaABF7 and 8 correspond to the same gene model in the latest version of the maize genome. Although there are several orthologues of this gene ID in other species, the sequence it encodes is too short for a functional product. As a result, we excluded this gene from further analysis.
I'm still wondering if ZmaABF3, ZmaABF6 and ZmaABF7-8 are pseudogenes or not?
Author Response
Dear reviewer,
Thank you for your valuable comments.
We have updated the statistics on the frequency of CBM4-like domain ocuurence in GH51 proteins of plants, fungi, and bacteria. We have also added an information on insect proteins. The Materials and Methods section has also been modified accordingly.
ZmaABF3 is a pseudogene (this is how it is defined in the NCBI database, but no specific reason is given). ZmaABF6, 7, and 8 showed the highest similarity to the same gene of a newer assembly. However, a search for the coding gene using the amino acid sequence of ZmaABF6 yielded no results, whereas the protein sequences of ZmaABF7 and 8 both corresponded to LOC109944448. Thus, ZmaABF7 and 8 were probably merged into a single gene model.
For your convenience, a version of the article with traced changes has been added to the supplementary material.
Respectfully,
Alsu Nazipova and co-authors
Reviewer 3 Report
All of my previously raised issues have been addressed appropriately.